# GWAS and 3D chromatin mapping identifies multicancer risk genes associated with hormone-dependent cancers

Isela Sarahi Rivera[1,2], Juliet D. French[1,2,3]☯*, Mainá Bitar[1,3], Haran Sivakumaran[1], Sneha Nair[1], Susanne Kaufmann[1], Kristine M. Hillman[1], Mahdi Moradi Marjaneh[1,4], Jonathan Beesley[1,3]☯, Stacey L. Edwards[1,2,3]☯

**1** Cancer Research Program, QIMR Berghofer Medical Research Institute, Brisbane, Australia, **2** Faculty of Health, Queensland University of Technology, Brisbane, Australia, **3** Faculty of Medicine, The University of Queensland, Brisbane, Australia, **4** Department of Infectious Disease, Imperial College London, London, United Kingdom

☯ These authors contributed equally to this work.
* juliet.french@qimrberghofer.edu.au

**Data Availability Statement:** The authors confirm that all data underlying the findings are fully available without restriction. The PCHiC data generated as part of this study are accessible in the

## Abstract

Hormone-dependent cancers (HDCs) share several risk factors, suggesting a common aetiology. Using data from genome-wide association studies, we showed spatial clustering of risk variants across four HDCs (breast, endometrial, ovarian and prostate cancers), contrasting with genetically uncorrelated traits. We identified 44 multi-HDC risk regions across the genome, defined as overlapping risk regions for at least two HDCs: two regions contained risk variants for all four HDCs, 13 for three HDCs and 28 for two HDCs. Integrating GWAS data, epigenomic profiling and promoter capture HiC maps from diverse cell line models, we annotated 53 candidate risk genes at 22 multi-HDC risk regions. These targets were enriched for established genes from the COSMIC Cancer Gene Census, but many had no previously reported pleiotropic roles. Additionally, we pinpointed lncRNAs as potential HDC targets and identified risk alleles in several regions that altered transcription factors motifs, suggesting regulatory mechanisms. Known drug targets were over-represented among the candidate multi-HDC risk genes, implying that some may serve as targets for therapeutic development or facilitate the repurposing of existing treatments for HDC. Our approach provides a framework for identifying common target genes driving complex traits and enhances understanding of HDC susceptibility.

## Author summary

While hormone-dependent cancers (HDCs) share several risk factors, our understanding of the complex genetic interactions contributing to their development is limited. In this study, we leveraged large-scale genetic studies of cancer risk, high-throughput sequencing methods and computational analyses to identify genes associated with four HDCs: breast, endometrial, ovarian and prostate cancers. We identified known cancer genes and discovered many that were not previously linked to cancer. These findings are significant

Gene Expression Omnibus database under the accession GSE278254 (https://www.ncbi.nlm.nih.gov/geo/query/acc.cgi?acc=GSE278254). The published results data were retrieved through database of Genotypes and Phenotypes (dbGaP) authorization (accession number:phs000178.v11.p8).

**Funding:** This work was funded by a grant from the Cancer Council Queensland (1156712; to JDF and SLE). Funding by the QIMR Berghofer Maureen Stevenson PhD Scholarship, a QIMR Berghofer top-up scholarship and a QUT HDR Tuition fee sponsorship covered the salary of I.S.R. A philanthropic donation from Isabel and Roderic Allpass and an NHMRC Investigator Grant covered the salary of J.D.F (2016826). An NHMRC Senior Research Fellowship covered the salary of S.L.E (1135932). Funding for open access charge: QIMR Berghofer offers institutional grants. The funders had no role in study design, data collection and analysis, decision to publish, or preparation of the manuscript.

**Competing interests:** The authors have declared that no competing interests exist.

because identifying genes associated with risk of multiple cancer types can enhance the gene mapping accuracy and highlight new therapeutic targets.

## Introduction

Breast, endometrial, ovarian and prostate cancers are hormone-dependent cancers (HDCs) that together account for up to 30% of new cancer diagnoses each year [1]. These cancers share several environmental, behavioural and genetic risk factors, suggesting a common aetiology [2,3]. This premise is supported by genome-wide association studies (GWAS) which have identified hundreds of cancer-specific risk loci [4–7] and multiple pleiotropic loci associated with at least two HDCs [8–11]. The detection of pleiotropic loci suggests that shared genetic factors likely contribute to polygenic risk of HDCs and raises the possibility of common driver genes and biological pathways.

A key aim of post-GWAS is to identify the target gene(s) that are affected at each GWAS region. This is complicated by the fact that most risk variants reside in noncoding regions of the genome making it difficult to interpret how they contribute to cancer susceptibility [12]. Target gene mapping for pleiotropic loci typically relies on statistical approaches such as expression quantitative trait loci (eQTL) and transcriptome-wide association studies (TWAS) [9,10,13]. However, these methods are limited by small sample sizes which reduces power, the use of steady-state gene expression and the lack of data from relevant tissues. Orthologous functional assays provide complementary mapping approaches to better define regulatory variants and connect them to their target genes.

## Results and discussion

While most GWAS variants are dispersed across the genome, several genomic regions harbour variants for multiple cancer types, suggesting common target gene(s) may drive these associations. Independent signals for HDC were obtained from large-scale GWAS, meta-analyses and fine-mapping (196 breast [4], 17 endometrial [5], 60 ovarian [6] and 258 prostate cancer index variants [7]). To assess whether HDC variants were more frequently positioned together than would be expected by chance, we defined "clusters" when variants associated with two or more HDCs were co-localized within the same 100 kb window. We compared observed cluster frequency to a null distribution, generated by randomly shuffling variants one million times (see Methods), and recounted cluster occurrence in each random set. This resulted in a statistically significant increase in observed frequency (**Fig 1A**; $P < 10^{-6}$, permutation test). As an additional comparison, index variants for four genetically uncorrelated traits [14] were obtained from the Open Target Genetics portal (179 for coronary artery disease, 22 lung cancer, 38 Alzheimer's disease and 94 Parkinson's disease [15]). These traits showed similar genomic distributions to randomly generated background regions (**Fig 1A**), which provided support that HDC risk regions co-localize in the genome and that these regions likely contain pleiotropic variants and common target genes.

To identify candidate multi-HDC risk (mHDCR) regions, we extended the variants of each HDC by 0.5 Mb on either side, resulting in cancer-specific risk regions. mHDCR regions were defined as areas where the cancer-specific risk regions of at least two HDCs overlapped (**Fig 1B**). We did not consider linkage disequilibrium (LD) between variants of each cancer type, focussing solely on their positional relationships. In total, 44 mHDCR regions were identified across the genome (**Fig 1C** and **S1 Table**). Two mHDCR regions contained risk variants for all four HDCs, 13 mHDCR regions for three HDCs (seven breast-ovarian-prostate, four

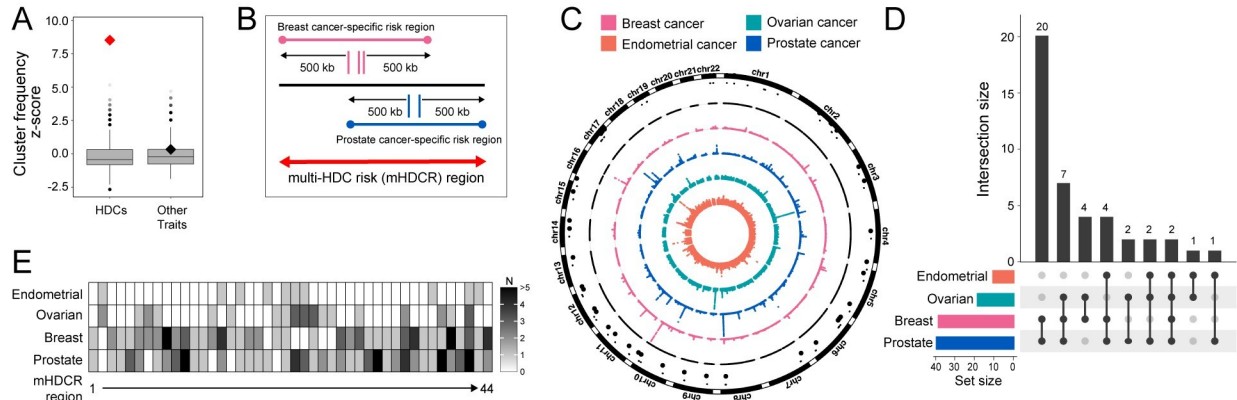

**Fig 1. GWAS variants from four HDCs cluster in mHDCR regions. (A).** Boxplots show the null distribution of z-scores from $10^6$ random permutations of the positions of variants associated with four HDCs (breast, endometrial, ovarian and prostate cancers), and four genetically uncorrelated traits (coronary artery disease, lung cancer, Alzheimer's disease and Parkinson's disease). The true cluster count score is indicated with a diamond, boxes represent the median and interquartile range and whiskers represent upper and lower quartiles of the null distribution. **(B).** Schematic of a hypothetical mHDCR region (red arrow) containing variants for breast cancer (pink) and prostate cancer (blue) which were extended to generate the cancer-specific risk regions. **(C).** Circus plot providing an overview of the association results. Dots in the four coloured rings correspond to genome-wide significant ($-\log_{10}(5 \times 10^{-8})$) GWAS variants identified for breast, endometrial, ovarian and prostate cancers, ordered by chromosomal position. The black ring denotes chromosomes (chr) 1–22 and the black dots specify the 44 mHDCR regions. **(D).** Upset plot to illustrate the number of mHDCR regions shared between the four HDCs. The vertical barplot represents the total number of mHDCR regions in each HDC combination. Points and lines in the matrix visualize these connections, and the colored horizontal bars are the total number of mHDCRs in each HDC. **(E).** Heatmap showing the number (N) of independent GWAS signals in each mHDCR region per cancer type.

breast-endometrial-prostate and two endometrial-ovarian-prostate) and 28 mHDCR regions for two cancer types (twenty breast-prostate, four breast-ovarian, two ovarian-prostate, one endometrial-ovarian and one endometrial-prostate) (**Fig 1D**). Compared to ovarian and endometrial cancers, the significantly larger sample sizes [4, 7] resulted in most breast and prostate mHDCR regions containing at least three independent signals per cancer type (**Fig 1E**).

Candidate mHDCR gene(s) within the 44 mHDCR regions were identified using a multi-step computational approach (**Fig 2A**). For endometrial, ovarian and prostate cancers, each variant was expanded to include all candidate variants ($r^2 \geq 0.8$; 1000 Genomes phase 3 version 5 reference) (**S2 Table**). For breast cancer, the candidate causal variants from the recent fine-mapping study were used [4]. We first annotated each mHDCR region with protein-coding genes from GENCODE (v37 basic) and intersected all candidate variants with genomic annotations (exons, promoters defined as transcription start site (TSS) ± 2 kb and intronic and intergenic regions; **S1A Fig**). In total, 134 (~6%) of the risk variants mapped to protein-coding gene exons, while 4132 (~94%) were located in noncoding regions. Exonic variants were filtered using Ensembl variant effect predictor (VEP) [16] which identified one frameshift and 33 moderate-impact missense and untranslated region (UTR) variants (**S1B Fig** and **S3 Table**). Seven potential splicing variants were also detected with SpliceAI [17] and MaxEntScan [18]. The noncoding variants were further explored for regulatory functions using RegulomeDB [19] and ChromHMM data from EpiMap [20]. Pre-calculated RegulomeDB scores for dbSNP v153 common SNPs (MAF ≥ 0.01) were downloaded from regulomedb.org. From this dataset, we extracted two key metrics: the RegulomeDB ranking, where a score of 1 indicates stronger supporting evidence for regulatory function and probability scores, where a score of 1 indicates higher likelihood that a variant is regulatory (**S4 Table**). ChromHMM data was available for three breast, five prostate, three ovarian and one endometrial tissue sample or cell line. We annotated the chromatin states of all potential regulatory variants using ChromHMM

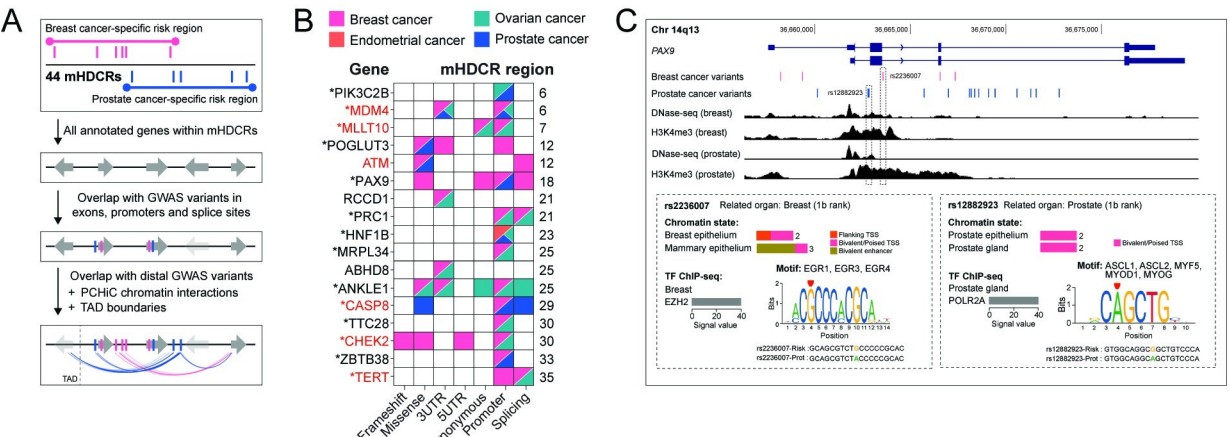

**Fig 2. Identification of candidate mHDCR genes. (A).** Schematic of the stepwise computational pipeline used to identify candidate mHDCR genes. **(B).** Summary of candidate mHDCR genes after overlap with variants in gene exons, promoters and splice sites that are associated with two or more HDCs (pink-breast cancer; orange-endometrial cancer; teal-ovarian cancer; blue-prostate cancer). The left y-axis shows the annotated gene names, the right y-axis denotes the mHDCR region. The asterisk denotes genes with evidence of cell-type-specific regulatory activity from ChromHMM. The red text highlights known cancer genes (from COSMIC) at the mHDCR regions. The x-axis provides the location and/or functional annotation of the HDC risk variants. Frameshift, missense, synonymous-variants located in gene exons; 3'UTR/5'UTR-variants; Promoter-variants located in gene promoters defined as TSS ± 2 kb; Splicing-variants in gene introns that are predicted to alter splicing. **(C).** WashU genome browser (hg38) showing GENCODE annotated genes (blue). The risk variants are shown as pink (breast cancer) and blue (prostate cancer) vertical lines. The DNase-seq and H3K4me3 tracks from breast and prostate cells are shown as black histograms. The dashed gray outlines highlight the functional variants. Insets: RegulomeDB v2.2 analysis for rs2236007 and rs12882923 including heuristic ranking scores, presence in promoters or enhancers (chromatin state) and sequences affecting the binding of TFs (TF ChIP-seq) and DNA motifs.

signatures [20] (S4 Table). Previous studies have performed functional assays for some of the noncoding variants in individual HDCs [21–24]. For example, Lawrenson et al used 3D mapping and reporter assays to show that breast and ovarian cancer risk variants at 19p13 influence distal enhancers, which in turn regulate *ABHD8* expression [22]. Furthermore, Stegeman et al used reporter assays to show a prostate cancer risk variant alters microRNA binding to the *MDM4* 3'UTR [23]. These studies provide additional support that the variants can impact target gene expression through various mechanisms.

An initial analysis identified 17 candidate mHDCR genes (across 11 mHDCR regions) that contain exonic, promoter or potential splicing variants associated with two or more HDCs (Figs 2B and S1C and S5 Table). For those candidate mHDCR genes with promoter variants, all were supported by evidence of cell-type-specific regulatory activity from ChromHMM (S4 Table). Additionally, six are cancer genes based on information from the Catalogue of Somatic Mutations in Cancer (COSMIC [25]), while the remaining genes may represent new pleiotropic HDC risk genes. One example is the *PAX9* transcription factor (TF; mHDCR region 18) [26] which has a missense breast cancer risk variant plus promoter and intronic breast and prostate cancer risk variants (Fig 2C). Rs2236007 (breast) and rs12882923 (prostate) were the top-ranked variants on RegulomeDB and mapped to transcriptionally active open chromatin (ChromHMM TSS signatures in breast and prostate samples; S4 Table). Previous studies in breast cells showed that the rs2236007 risk *g*-allele reduced *PAX9* promoter activity [27], via recruitment of the suppressive TF EGR1 and increased breast cancer cell growth [28]. Furthermore, rs12882923 maps to a POL2RA binding site and the DNA motif of multiple TFs in prostate epithelium (Fig 2C), suggesting the variant has functional impact.

Most of the identified HDC risk variants are noncoding (S1A Fig; ~94%), and a subset of these will map to DNA regulatory elements that modulate gene transcription through long-

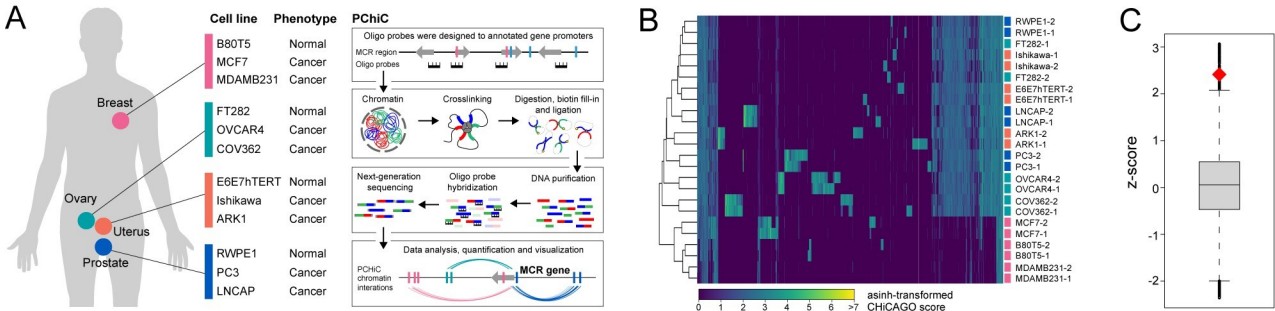

**Fig 3. PCHiC in HDC cell lines. (A).** Schematic of PCHiC experimental approach. **(B).** Agglomerative hierarchical clustering for the PCHiC in the twelve cell lines. **(C).** Enrichment of PCHiC interactions within TADs. The boxplot shows the null distribution z-scores generated from randomly permuting the TAD boundary positions within mHDCR regions (n = $10^5$) and counting the number of interactions which fall within randomized TAD boundaries. The diamond indicates the z-score of the count of interactions which fall within observed TAD boundaries. The box represents the median and interquartile range, whiskers represent upper and lower quartiles of the null distribution.

range chromatin interactions [29]. To compile a comprehensive list of candidate mHDCR genes, we next intersected the intronic and intergenic HDC variants with the 44 mHDCR regions. Promoter capture HiC (PCHiC) was used to assign distal regulatory variants to their target gene promoters (**Fig 3A**). PCHiC probes were designed to 2774 HindIII fragments containing 3096 GENCODE gene promoters that fall within the 44 mHDCR regions (**S6 Table**). PCHiC data was derived from twelve breast, endometrial, ovarian and prostate non-tumorigenic and cancer cell lines, as described previously [30] (**S2A Fig** and **S7 Table**). Using an interaction score threshold (CHiCAGO score ≥ 5) we detected 8–19,000 high-confidence interactions per cell type across the 44 mHDCR regions with strong correlation between replicates (**Figs 3B** and **S2B**). We prioritised the PCHiC interactions based on topologically associating domain (TAD) boundaries [31]. To show that a published set of TAD annotations marked regions of increased regulatory activity, we examined the relationship between our PCHiC interactions and TAD boundaries. We observed a significant enrichment of PCHiC interactions within TAD boundaries, compared to randomly placed TAD boundaries ($P$ = 0.0297, permutation test), suggesting that the published TAD boundaries mark the limits of most PCHiC interactions in our HDC cell lines (**Fig 3C**).

The combination of variant intersection, PCHiC and TAD boundaries identified 53 candidate mHDCR genes (at 22 mHDCR regions) associated with two or more HDCs (**Fig 4A** and **S5 Table**). The majority of candidate mHDCR genes (45/53) were associated with breast and/ or prostate cancers, which is likely attributable to the larger GWAS sample sizes for these cancers. Ten candidate mHDCR genes were associated with three cancer types (seven breast-ovarian-prostate and three endometrial-ovarian-prostate) and 43 candidate mHDCR genes with two cancer types (seventeen breast-ovarian, fifteen breast-prostate, eight endometrial-prostate and three ovarian-prostate) (**Fig 4B**). The expression of all candidate mHDCR genes in the relevant tissue or cancer type was assessed using GTEx and TCGA RNA-seq data (**S3 Fig**). Twenty-five genes had evidence of cell-type-specific regulatory activity based on ChromHMM data from the corresponding cell types (**Fig 4A** and **S5 Table**), and 19 were supported by evidence from a single cancer type (**S5 Table**). The remaining genes were retained as mHDCR candidates, as their regulatory activity may be context-dependent or associated with specific cell or cancer subtypes. Twelve are cancer genes from COSMIC, doubling the number of candidate HDC cancer genes when regulatory variants are taken into account (**Fig 4A**). Of these, nine are supported by cell-type-specific regulatory activity based on ChromHMM. As one example, *TCF7L2* (transcription factor 7 like 2) has 5'UTR, promoter and intronic prostate

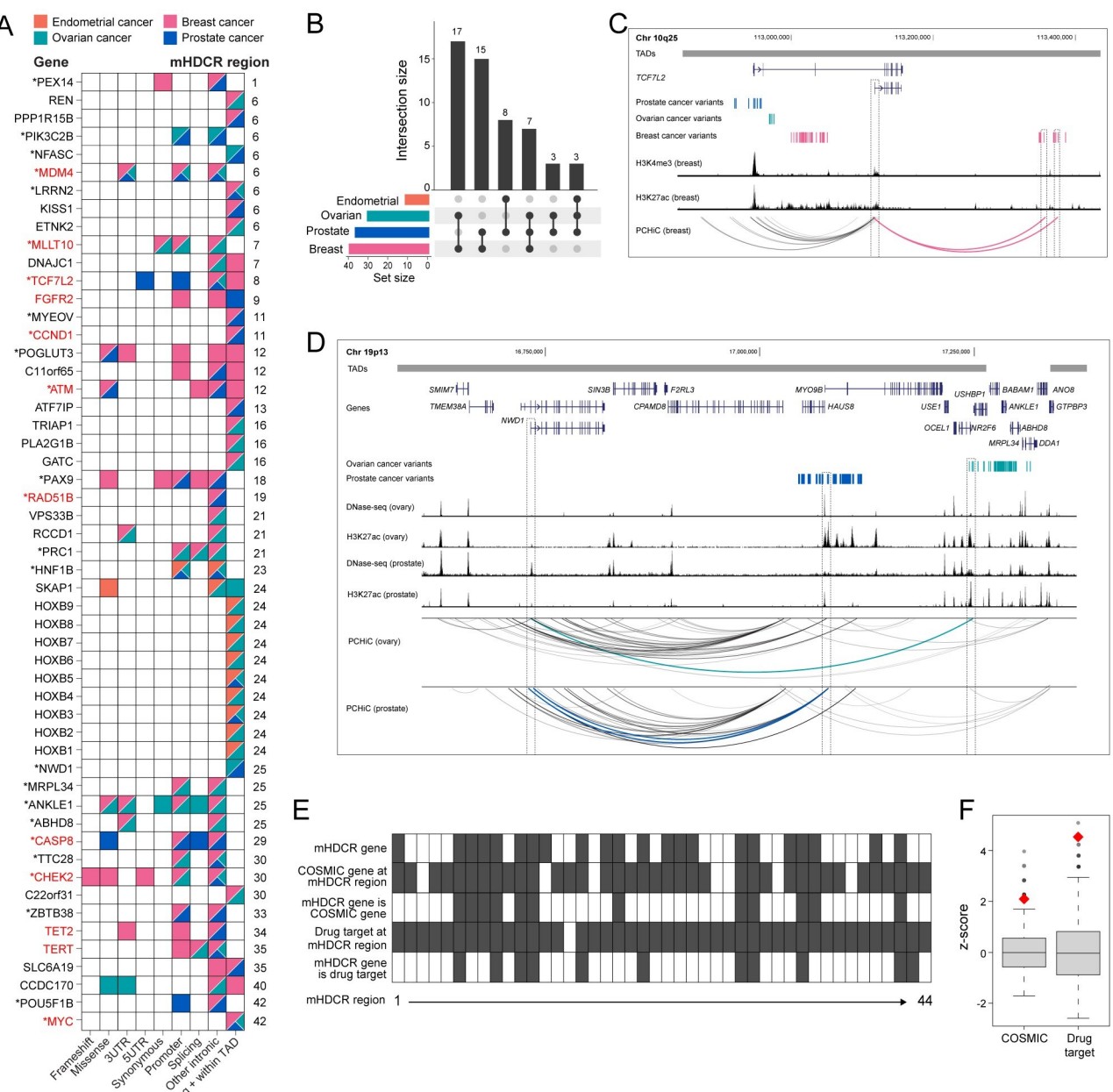

**Fig 4. Annotation of candidate mHDCR genes. (A).** Summary of all candidate mHDCR genes that are associated with two or more HDCs (pink-breast cancer; orange-endometrial cancer; teal-ovarian cancer; blue-prostate cancer). The left y-axis shows the annotated gene names, the right y-axis denotes the mHDCR region. The asterisk denotes genes with evidence of cell-type-specific regulatory activity from ChromHMM. The red text highlights known cancer genes (from COSMIC) at the mHDCR regions. The x-axis provides the location and/or functional annotation of the HDC risk variants. Frameshift, missense, synonymous-variants located in gene exons; 3'UTR/5'UTR-variants; Promoter-variants located in gene promoters defined as TSS ± 2 kb; Splicing-variants in gene introns that are predicted to alter splicing; Other intronic-variants located in gene introns and RegulomeDB scores indicate the variants are regulatory; Reg + within TAD-variants located outside any genes, RegulomeDB scores indicate the variants are regulatory, our PCHiC data show chromatin interactions between the variant and gene promoters and the interactions are within defined TAD boundaries. **(B).** Upset plot to illustrate the number of candidate mHDCR genes shared between the four HDCs. The vertical barplot represents the total number of candidate mHDCR genes in each HDC combination. Points and lines in the matrix visualize these connections, and the colored horizontal bars are the total number of candidate mHDCR genes in each HDC. **(C).** WashU genome browser (hg38) showing TADs as horizontal gray bars above GENCODE genes (blue). The risk variants are shown as blue (prostate cancer), teal (ovarian cancer) and pink (breast cancer) vertical lines. The H3K4me3 and H3K27ac tracks from breast cells are shown as black histograms. CHiCAGO-scored interactions are shown as colored arcs. The dashed gray outlines highlight distal breast cancer variants and the target gene (*TCF7L2*). **(D).** WashU genome browser (hg38) showing TADs as horizontal gray bars above GENCODE genes (blue). The risk variants are shown as teal (ovarian cancer) and blue (prostate cancer) vertical lines. The DNase-seq and H3K27ac tracks from ovary and prostate cells are shown as black histograms. CHiCAGO-scored interactions are shown as colored

arcs. The dashed gray outlines highlight the likely functional variants and the target gene (*NWD1*). **(E).** Summary of identified candidate mHDCR genes at the 44 mHDCR regions for HDCs. Filled boxes represent presence of the feature according to the row title. **(F).** Enrichment of COSMIC genes and genes which encode known drug targets in candidate mHDCR genes. The boxplots represent the null distribution z-scores following $10^5$ random permutations of category labels (COSMIC or drug target) of genes at mHDCR regions, then recounting the number of predicted mHDCR genes belonging to the category. The z-scores for the true observed counts are shown as red diamonds. Boxes represent the median and interquartile range and whiskers represent upper and lower quartiles of the null distribution.

cancer variants that are associated with the risk of developing aggressive prostate cancer [32] (**Fig 4C**). However, our combined analysis showed that *TCF7L2* also has intronic breast and ovarian cancer risk variants plus distal breast cancer risk variants which loop to and may regulate an alternate *TCF7L2* isoform (**Fig 4C**).

Of note, two-thirds of the genes in the final list were identified as candidate mHDCR genes only after incorporating distal variants and PCHiC data. Some are known pleiotropic cancer genes (e.g *MYC*, *CCND1*; **S4 Fig**) supporting the validity of our approach, but many have only been statistically and/or functionally associated with individual HDCs and may represent new mHDCR genes. One example is *NWD1* (NACHT and WD repeat domain containing 1) at 19p13 (**Fig 4D**). Previous studies have detected variants at this region associated with breast and ovarian cancer risk and identified *ABHD8* and *ANKLE* as the likely target genes [22]. Here we show that *NWD1*, located ~580 kb from *ABHD8*, has distal ovarian and prostate cancer variants, some of which fall in regulatory elements that loop to and may regulate the *NWD1* promoter (**Fig 4D**). Overexpression of NWD1 is reported to promote prostate tumor progression by modulation of androgen receptor (AR) signaling [33]. There is limited information about NWD1 function in ovarian cancer, but evidence suggests AR signaling also contributes to initiation and progression of this disease [34], providing a potential mechanism.

It is established that drug targets with genetic support (such as well-powered GWAS) are twice as likely to lead to approved drugs than those without such evidence [35]. To determine whether the candidate mHDCR genes were enriched for known drug targets, we mined the Citeline Pharmaprojects database. Pharmacological target data were sourced from a recent study [36]. To evaluate the significance of overlaps between known drug targets and candidate mHDCR genes, we applied Fisher's exact test, using a background set consisting of all candidate genes across all mHDCRs. For visualization of the enrichment, we compared the number of known drug targets to randomly sampled gene sets, matched to the number of genes per mHDCR region, performing 10,000 iterations and z-scaling the background distribution. A similar approach was used to assess the overlap between mHDCR genes and genes from the COSMIC. We found twelve candidate mHDCR genes that encode known preclinical, clinical phase and approved drug targets for various diseases (**Fig 4E** and **S8 Table**). The overlap between known drug targets and the candidate mHDCR genes was statistically significant (**Fig 4F**; OR = 2.2, *P* = 0.028), indicating the value of using our pipeline to identify candidates for drug repositioning.

Our approach was unable to identify candidate mHDCR genes at 22 regions. One possible explanation is that individual HDCs are driven by distinct genes. In fact, we identified over 300 genes associated with individual HDCs across 37 regions (**S9 Table**). However, it is important to note that this list is not exhaustive due to the limitations of our targeted approach. Furthermore, we focused on identifying protein-coding genes, but studies by ourselves (and others) show that long noncoding RNAs (lncRNAs) are also transcribed from cancer risk regions and can have important roles in tumorigenesis [37–41]. Indeed, when we intersected the HDC risk variants with annotated lncRNAs at the 44 mHDCR regions, we identified 21 lncRNAs associated with two or more HDCs, including two cancer-related lncRNAs that were the only mHDCR targets identified at regions 27 and 32 (**S5 Fig**) [42, 43]. Given most cell-type

specific lncRNAs are not annotated in current databases, we expect that additional novel lncRNAs will contribute to risk at other mHDCR regions. We also acknowledge that using a 6-base pair (bp) cutter in PCHiC reduces resolution and introduces specific biases compared to more recent methods employing 4-bp cutters. Additionally, the lack of suitable baits for selected promoters, the lower resolution for short-range interactions, and the transient, cell type-specific nature of chromatin interactions may contribute to false negatives. It is also important to note that interactions between risk variants and gene promoters do not infer causality and that follow-up studies are required to interpret GWAS results.

In summary, our study expands the repertoire of risk regions and candidate risk genes for HDCs, providing further insights into the complex genetic architecture and biology underpinning HDC susceptibility. The candidate mHDCR gene list was enriched for known cancer genes and drug targets, which provides support that other, less-well-characterized genes at mHDCR regions may play important roles in HDC development. Our approach highlights the value of performing integrative analyses of genetics and functional genomics to enhance pleiotropic cancer gene identification. Combined with future research that investigates functional mechanisms, our results may serve to redirect efforts to more promising targets for new drugs or allow drug repurposing for HDC treatment.

## Materials and methods

### Genomic interval operations

Genomic interval analyses were performed with BEDTools version 2.29.0 [44]. To assess the genomic distribution of genetic signals, the *cluster* sub-command was used to assign IDs to GWAS variants overlapping 100 kb windows. Co-located variants for two or more traits were designated as clusters. For comparison, positions of the mHDCR variants were shuffled while maintaining chromosomal distribution using BEDTools *shuffle* sub-command, and clusters counted for each permutation. To explore the relationship between PCHiC loops and TADs, boundary positions within multi-cancer regions were overlapped with loops using the BEDTools *intersect* sub-command. For comparison, we generated null distributions by shuffling TAD boundary positions within mHDCRs and repeating 10,000 times. TAD size was maintained by circularising the mHDCR, such that if a TAD boundary was randomly placed beyond the mHDCR limits, it was moved to the start of the mHDCR. The background count was performed by intersecting loops with each iteration of randomly permuted boundary sets. The significance of these tests were defined by the number of times the count from random permutations was greater or equal to the observed overlap, divided by total number of permutations. For presentation, the null distribution was standardised to produce z-scores and shown as box plots.

### Variant annotation

Candidate causal variant rsIDs were submitted to the Variant Effect Predictor (VEP) web interface. Coding variants annotated as frameshift, missense, 3'UTR, 5'UTR or synonymous were prioritized as potentially functional with an impact on the associated transcript. Intronic variants with the potential to alter splicing were identified using 1) SpliceAI Delta scores $\geq 0.2$, where delta can be interpreted as the probability that the variant is splice altering; or 2) absolute MaxEntScan maximum entropy scores $> 0$. The regulatory effects of all variants were assessed using RegulomeDB. Variants with a probability score greater than the median (0.55) were considered as potential functional. Chromatin HMM data were downloaded from the EpiMap Repository (https://compbio.mit.edu/epimap/). Genomic coordinates for topologically associating domains (TADs) were obtained from a recent publication [45]. Annotation of

variants with respect to genomic intervals (exons, introns, TADs, and promoters) and to link with candidate distal target genes were performed with the GenomicRanges and GenomicInteractions BioConductor packages. Genes with known roles in cancer were obtained from the COSMIC Cancer Gene Census, version 99 [25]. Information about drug targets and indications was obtained from [36].

## Cell lines and culture conditions

MCF7, MDAMB231, FT282, RWPE1, LNCAP and PC3 cells were purchased from ATCC. OVCAR4 and COV362 were purchased from Sigma-Aldrich. B80T5 cells were a gift from Prof R Reddel (CMRI, Australia). Ishikawa and E6E7hTERT cells were a gift from Prof PM Pollock (QUT, Australia). ARK1 cells were provided by Prof AD Santin (UAMS, USA). MCF7 cells were cultured in RPMI medium with 10% (vol/vol) fetal bovine serum (FBS, Gibco), 1% (vol/vol) antibiotic/antimycotic (a/a, Gibco), 1mM sodium pyruvate (Gibco), 0.02M Hepes (pH 7.0–7.4) and 10 μg/mL human recombinant insulin (Gibco). MDA-MB-231 cells were grown in RPMI medium with 10% (vol/vol) FBS, 1% (vol/vol) a/a, 1 mM sodium pyruvate, 0.02M Hepes (pH 7.0–7.4). B80T5, ARK1, LNCAP and OVCAR4 cells were grown in RPMI medium with 10% (vol/vol) FBS and 1% (vol/vol) a/a. E6E7hTERT, Ishikawa and COV362 cells were grown in DMEM medium with 10% (vol/vol) FBS and 1% (vol/vol) a/a. PC3 and FT282 cells were grown in DMEM:F12 medium with 10% (vol/vol) FBS and 1% (vol/vol) a/a. RWPE1 cells were grown in Gibco Keratinocyte-SFM Combo (Cat# 17005042) with 1% (vol/vol) a/a. Cell lines were maintained at 37°C and 5% $CO_2$, routinely tested for *Mycoplasma* and authenticated using short tandem repeats (STR) profiling.

## Biotinylated RNA bait library design and HiC libraries capture

To generate target regions for PCHiC, mHDCR regions were extended by 1Mb on either side, so that all possible promoter-enhancer interactions involving regions containing HDC risk variants were captured. All HindIII fragments containing annotated promoters within each extended region were identified. HindIII fragments previously captured by Javierre *et al* [46] were labelled 'group1' and those not captured but overlapping Ensembl annotated promoters [47] were labelled 'group2'. Capture probes were designed to both ends of group1 and group2 HindIII fragments (2783 fragments and 4023 probes, respectively).

## Promoter capture HiC (PCHiC) library preparation and sequencing

PCHiC was performed on nine immortalized HDC cell lines. For prostate: one normal cell line (RWPE1), one androgen-dependent (LNCAP) and one castration-resistant (PC3) cell line. For ovarian: one normal ovarian cancer precursor cell line (FT282) and two high-grade serous ovarian cancer cell lines (OVCAR4 and COV362). For endometrial: one normal endometrial cell line (E6E7hTERT), one ER+ (Ishikawa) and one type II EC line (ARK1). For breast: we remapped our published PCHiC data to mHDCR regions [29]. PCHiC libraries were prepared from 4–5 x $10^7$ cells per library (two biological replicates per cell line using in-nucleus ligation [48]). The HiC libraries were amplified using the SureSelectXT ILM Indexing pre-capture primers (Agilent Technologies) with 8 PCR amplification cycles. Each HiC library (750 ng) was hybridized and captured individually using the SureSelectXT Target Enrichment System reagents and protocol (Agilent Technologies). After library enrichment, a post-capture PCR amplification step was carried out using SureSelectXT ILM Indexing post-capture primers (Agilent Technologies) with 14–16 PCR amplification cycles. PCHiC libraries were multiplexed and sequenced on NovoSeq 6000 S4 (Novogene, Singapore).

## PCHiC analysis

PCHiC reads were analysed with *HiCUP* (version 0.5.9). Raw sequencing reads were aligned to the hg38 human reference genome with *Bowtie2* version 2.2.9 and filtered to remove experimental artefacts (e.g. re-ligated, circularized, unligated fragments or fragments separated by less than 20 kb). Chromatin interactions were identified using CHiCAGO following generation of genome indices, HindIII restriction fragment digest files, and the bait map. The CHiCAGO pipeline was run independently for each library for quality control, followed by merging of replicates for subsequent analyses. Interactions with CHiCAGO score $\geq$ 5 were treated as high confidence for target gene analysis and visualisation.

## Supporting information

**S1 Fig. Annotation of GWAS variants and target genes at mHDCR regions. (A).** The numbers and percentage of risk variants located in exons, promoters (TSS ± 2 kb), introns and intergenic regions. **(B).** The percentage of exonic risk variants in different functional categories. **(C).** Schematic representation of the 17 candidate mHDCR genes that contain exonic, promoter or potential splicing variants associated with two or more HDCs. WashU genome browser (hg38) showing GENCODE annotated genes (blue) and risk variants as coloured vertical lines (pink-breast cancer; orange-endometrial cancer; teal-ovarian cancer; blue-prostate cancer).
(TIF)

**S2 Fig. PCHiC CHiCAGO-identified interaction characteristics. (A).** Scatter plots showing the correlation between duplicate PCHiC libraries based on the number of raw di-tags mapping to interaction fragment pairs. ρ is Spearman's correlation. **(B).** Principal component analysis (PCA) of CHiCAGO-scored interactions in PCHiC biological replicates for breast cells lines (top panel) or prostate, ovarian and endometrial cell lines (bottom panel).
(TIF)

**S3 Fig. Candidate mHDCR gene expression.** Heatmap showing candidate mHDCR gene expression from normal tissue samples in GTEx and tumor samples in TCGA.
(TIF)

**S4 Fig. Candidate mHDCR genes. (A).** WashU genome browser (hg38) showing TADs as horizontal gray bars above GENCODE genes (blue). The risk variants are shown as pink (breast cancer), teal (ovarian cancer) and blue (prostate cancer) vertical lines. The H3K27ac tracks from breast, ovary and prostate cells are shown as black histograms. CHiCAGO-scored interactions are shown as colored arcs. The dashed gray outline highlights the target gene (*MYC*). **(B).** WashU genome browser (hg38) showing TADs as horizontal gray bars above GENCODE genes (blue). The risk variants are shown as pink (breast cancer) and blue (prostate cancer) vertical lines. The H3K27ac tracks from breast and prostate cells are shown as black histograms. CHiCAGO-scored interactions are shown as colored arcs. The dashed gray outline highlights the target gene (*CCND1*).
(TIF)

**S5 Fig. Identification of candidate mHDCR long noncoding RNAs (lncRNAs).** Summary of candidate mHDCR lncRNAs that are associated with two or more HDCs (pink-breast cancer; orange-endometrial cancer; teal-ovarian cancer; blue-prostate cancer). The left y-axis shows the lncRNA names, the right y-axis denotes the mHDCR region. The asterisk denotes genes with evidence of cell-type-specific regulatory activity from ChromHMM. The red text highlights lncRNAs that are the only identified target at the mHDCR region. The x-axis provides

the location and/or functional annotation of the HDC risk variants. Exon-variants located in lncRNA exons; Promoter-variants located in lncRNA promoters defined as TSS ± 2 kb; Other intronic-variants located in lncRNA introns and RegulomeDB scores indicate the variants are regulatory; Reg + within TAD-variants located outside any lncRNA transcripts, RegulomeDB scores indicate the variants are regulatory, the PCHiC data show chromatin interactions between the variant and lncRNA promoters and the interactions are within defined topologically associated domain (TAD) boundaries.
(TIF)

**S1 Table. Identified mHDCR regions for breast, endometrial, ovarian and prostate cancers.**
(XLSX)

**S2 Table. GWAS variants (r2 ≥ 0.8; 1000G Phase3 v5 Reference) at the mHDCR regions.**
(XLSX)

**S3 Table. Variants predicted to be functional by VEP.**
(XLSX)

**S4 Table. Variants predicted to be regulatory by RegulomeDB.**
(XLSX)

**S5 Table. Candidate mHDCR genes.**
(XLSX)

**S6 Table. PCHiC baited HindIII fragments within mHDCR regions.**
(XLSX)

**S7 Table. Summary of the PCHiC CHiCAGO-scored interactions.**
(XLSX)

**S8 Table. Summary of genes at mHDCR regions annotated for mHDCR gene, COSMIC and drug targets.**
(XLSX)

**S9 Table. Genes associated with individual HDCs.**
(XLSX)

## Acknowledgments

The Genotype-Tissue Expression (GTEx) Project was supported by the Common Fund of the Office of the Director of the National Institutes of Health, and by NCI, NHGRI, NHLBI, NIDA, NIMH, and NINDS. The results published here are based in part upon data generated by The Cancer Genome Atlas (TCGA), managed by the NCI and NHGRI.

## Author Contributions

**Conceptualization:** Juliet D. French, Stacey L. Edwards.

**Formal analysis:** Isela Sarahi Rivera, Mainá Bitar, Mahdi Moradi Marjaneh, Jonathan Beesley.

**Funding acquisition:** Juliet D. French, Stacey L. Edwards.

**Investigation:** Haran Sivakumaran, Sneha Nair, Susanne Kaufmann, Kristine M. Hillman.

**Supervision:** Juliet D. French, Jonathan Beesley, Stacey L. Edwards.

**Writing – original draft:** Isela Sarahi Rivera, Juliet D. French, Jonathan Beesley, Stacey L. Edwards.

**Writing – review & editing:** Mainá Bitar, Haran Sivakumaran, Sneha Nair, Susanne Kaufmann, Kristine M. Hillman, Mahdi Moradi Marjaneh.

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
