## [Decision Letter · Decision Letter 0]

29 Aug 2024

Dear Dr Edwards,

Thank you very much for submitting your Research Article entitled 'GWAS and 3D chromatin mapping identifies multicancer risk genes associated with hormone-dependent cancers' to PLOS Genetics.

The manuscript was fully evaluated at the editorial level and by independent peer reviewers. The reviewers appreciated the attention to an important problem, but raised some substantial concerns about the current manuscript. Based on the reviews, we will not be able to accept this version of the manuscript, but we would be willing to review a much-revised version. We cannot, of course, promise publication at that time.

If you decide to revise the manuscript for further consideration at PLOS Genetics, please aim to resubmit within the next 60 days, unless it will take extra time to address the concerns of the reviewers, in which case we would appreciate an expected resubmission date by email to plosgenetics@plos.org.

To resubmit, log into your Editorial Manager account and select the option 'Revise Submission' in the 'Submissions Needing Revision' folder.

We are sorry that we cannot be more positive about your manuscript at this stage. Please do not hesitate to contact us if you have any concerns or questions.

Yours sincerely,

Gregory M. Cooper, PhD

Section Editor

PLOS Genetics

David Kwiatkowski

%CORR_ED_EDITOR_ROLE%

PLOS Genetics

Reviewer's Responses to Questions

**Comments to the Authors:**

Reviewer #1: In this paper, Rivera et al described GWAS together with 3D genome organization mapping in order to identify multicancer risk genes that are associated with hormone-dependent cancers such as prostate cancer, breast cancer, etc. This is a useful study, as it helps scientists identify multicancer risk genes that could be studied further for the development of new drugs, and so on. Overall, I found the study clear and easy to understand. I also appreciated the detailed Supplementary Tables with lists of targets. I recommend this study for acceptance in PLoS Genetics and do not have any further suggestions or comments.

Reviewer #2: It has been proposed that hormone-dependent cancers (HDCs) have a degree of common genetic etiology. The authors of this manuscript sought to determine the degree of such sharing and then implicate causal variants and genes at those loci shared across at least two traits. This is done with the integration of public domain and lab-generated datasets. Although the results represent an advance in the field, there are some concerns:

1. The authors state that "most HDC risk variants are noncoding" - it would be helpful for the reader to know this number within the main text. The authors then go on to say "the promoter and intronic variants were further explored for regulatory functions using RegulomeDB" - there needs to better description in this context; in particular, were the cell types driving these observations relevant to these traits? If not, then this would be of concern. And it doesn't appear public domain or in-house generated ATAC-seq datasets were leveraged, which would give unbiased insights into where potential regulatory regions exist in the given cell types based on open chromatin.

2. The authors refer to their chromatin conformation capture approach as high resolution. This reviewer would argue that the use of HindIII, a 6-cutter, is actually low resolution by today’s standards given there are now even commercially available options based on 4-cutter resolution. Indeed, at this low resolution used, there is concern that there is a lack of precision in the genes implicated.

3. The investigators use "breast, endometrial, ovarian and prostate non-tumorigenic and cancer cell lines". Given the approach is to understand susceptibility variants to the given diseases, it would seem more relevant to assess their interaction landscape before disease onset. The concern would be that looking in post-onset settings there is dysregulation in the interactions and thus assessment of the priming of the susceptibility is clouded.

Reviewer #3: The manuscript by Rivera et al explores a descriptive pipeline to identify genes associated with multiple hormone dependent cancers (HDCs). They use a genomic data to identify genes associated with genetic loci that are related to multiple HDCs and show that these are enriched in cancer genes as well as known drug targets. The manuscript is very clearly written, and the results are demonstrated very well. I have some further points that might be good to address:

1. When defining mHDC regions, the authors are merging two HDC variants if they are within a certain neighborhood. But if they are still independent/ in low LD, should they be not considered as separate HDC regions rather than merging into one mHDC regions? The downstream target gene analysis might still be valid if two low-LD variants, corresponding to different HDCs, have independent effects on the same gene.

2. I am curious as to what led the authors to favor investigating promoters as opposed to others like enhancers for which high quality variant-gene mappings are available? In general, there are a plethora of methods & scores that assign variants to genes based on their putative regulatory activity (e.g. work led by Alkes Price’s group and others), which possibly incorporated HiC data as well.

3. How was the overlap analysis for drug targets done? Why was one drug repository chosen? If drug repurposing is one of the final goals, I would imagine, looking at multiple drug databases would give a more robust result.

4. Given that the authors are looking for multi-HDC targets, analysis of GWAS with striking disparate power/sample size will skew the results towards discovering genes with effect on breast and/or prostate cancers.

5. Since identifying gene targets for multiple cancers is the theme, I would expect that the genes are enriched in hormone dependent processes and/or immunological processes. Can the authors investigate that?

6. In my opinion, although multi-HDC genes are important, the genes associated with single HDCs are also equally important, though slightly away from the overall goal, especially for the cancers which have lower power. Can the authors incorporate this in the discussion?

**Have all data underlying the figures and results presented in the manuscript been provided?**

Reviewer #1: Yes

Reviewer #2: Yes

Reviewer #3: Yes

PLOS authors have the option to publish the peer review history of their article (what does this mean?). If published, this will include your full peer review and any attached files.

Reviewer #1: **Yes: **Melissa Jane Fullwood

Reviewer #2: No

Reviewer #3: No

---

## [Decision Letter · Decision Letter 1]

6 Nov 2024

Dear Dr Edwards,

We are pleased to inform you that your manuscript entitled "GWAS and 3D chromatin mapping identifies multicancer risk genes associated with hormone-dependent cancers" has been editorially accepted for publication in PLOS Genetics. Congratulations!

Yours sincerely,

Gregory M. Cooper, PhD

Section Editor

PLOS Genetics

Gregory Cooper

Section Editor

PLOS Genetics

Aimée Dudley

Editor-in-Chief

PLOS Genetics

Anne Goriely

Editor-in-Chief

PLOS Genetics

Comments from the reviewers (if applicable):

Reviewer's Responses to Questions

**Comments to the Authors:**

Reviewer #3: The authors have addressed all my comments satisfactorily. No further comments from me.

**Have all data underlying the figures and results presented in the manuscript been provided?**

Reviewer #3: None

PLOS authors have the option to publish the peer review history of their article (what does this mean?). If published, this will include your full peer review and any attached files.

Reviewer #3: **Yes: **D Dutta

**Data Deposition**

http://datadryad.org/submit?journalID=pgenetics&manu=PGENETICS-D-24-00762R1

**Press Queries**

---

## [Editor Report · Acceptance letter]

19 Nov 2024

PGENETICS-D-24-00762R1 

GWAS and 3D chromatin mapping identifies multicancer risk genes associated with hormone-dependent cancers 

Dear Dr Edwards, 

We are pleased to inform you that your manuscript entitled "GWAS and 3D chromatin mapping identifies multicancer risk genes associated with hormone-dependent cancers" has been formally accepted for publication in PLOS Genetics! Your manuscript is now with our production department and you will be notified of the publication date in due course.

With kind regards,

Lilla Horvath

PLOS Genetics

On behalf of:
